# Polymerization in the Borstar Polypropylene Hybrid Process: Combining Technology and Catalyst for Optimized Product Performance

**DOI:** 10.3390/polym14214763

**Published:** 2022-11-07

**Authors:** Michiel F. Bergstra, Peter Denifl, Markus Gahleitner, Dusan Jeremic, Vasileios Kanellopoulos, Daniela Mileva, Pavel Shutov, Vasileios Touloupidis, Cornelia Tranninger

**Affiliations:** 1Borealis Polymers N.V., Industrieweg 148, 3580 Beringen, Belgium; 2Borealis Polyolefine GmbH, Innovation Headquarters, Sankt Peterstrasse 25, 4021 Linz, Austria

**Keywords:** polypropylene, catalyst, polymerization process, nucleation, application

## Abstract

Producing isotactic polypropylene (iPP) homo- and copolymers in a wide composition and property range according to customer demand requires perfect alignment between the process technology, catalyst system and polymer structure. The present review shows this for the Borstar^®^ PP process, a hybrid process employing liquid bulk and gas phase stages, in an exemplary way. It starts with the process design and continues through two generations of Ziegler–Natta catalyst development history to the design of advanced multimodal random and multiphase copolymers. Essential elements of each of the three areas contributing to performance range are highlighted, and an outlook to future development is given.

## 1. Introduction

The evolution of polypropylene (PP) from an experimental material in 1954 [1,2] to a material family, exhibiting a wide variety of possible microstructures, is impressive. Nowadays, polypropylene grades include multiple copolymers and compounds and attain a worldwide production volume of more than 60 million tons per year [3]. This success story has been enabled by the parallel development of catalyst technology and polymerization processes. The various steps along the way have not only made the production cheaper and less complex through a multifold increase in catalyst productivity, but also widened the product and property ranges.

Early first- and second-generation “Ziegler–Natta” (ZN) TiCl_3_ catalysts typically required the use of a hydrocarbon slurry and a series of multiple stirred-tank reactors to achieve acceptable productivity, with the reduction of atactic fraction being the main progress step between them. Nevertheless, catalyst productivity remained relatively low. For this reason, after polymerization, the polymer suspension had to be treated with alcohol to deactivate and solubilize the catalyst components; the suspension was then filtered to separate the residues and soluble fraction from the polymer. Products of this technology were limited to homopolymers with rather high molecular weight and a melt flow rate (MFR) 230 °C/2.16 kg below 15 g/10 min, random copolymers with low amounts of ethylene and impact-resistant copolymers of low amorphous phase content, with most limitations resulting from soluble fractions [4]. All following generations combine a MgCl_2_-based support with TiCl_4_ as an active centre. While the high-yield third-generation catalysts introduced by Montedison and Mitsui Petrochemical in the early 1970s [5,6] were limited in stereospecificity and still applied a high-yield slurry process, the introduction of aromatic monoesters as internal donors changed that situation and allowed the use of high-productivity bulk and gas-phase processes without atactic fraction removal [7].

In the 1980s, the use of phthalate esters as internal donors in combination with specifically designed and highly porous MgCl_2_ supports gave the fourth ZN catalyst generation unprecedented productivity levels and flexibility. Isotacticity was further increased by the use of alkoxysilanes as external donors. These catalysts can be employed both for liquid bulk and gas-phase polymerization processes [8,9]. Modern production units are either based on pure bulk polymerization in a loop reactor (LR) geometry, pure gas-phase polymerization in stirred (SBR) or fluidized bed reactors (FBR) or a combination of both, commonly called hybrid technology [10]. The “Spherizone” or multi-zone circulating gas-phase reactor (MZCR) technology of LyondellBasell [11] combines two distinct reaction zones with the smooth change of process conditions. The growing particle repeatedly circulates through these zones, which allows for the production of bimodal homo- and random PP products in one reactor. An overview of the most relevant technology schemes is given in Figure 1.

In recent years, increasing concerns about the negative health effects of phthalates [12] have increased the relevance of post-phthalate ZN catalysts. This has again broadened the field, as next to the first developments of LyondellBasell with diethers and succinates as internal donors [13,14], a wide range of alternatives is now being explored. An overview of the developments in the area of post-phthalate ZNCs can be found in the review by Severn [15]. In any case, the alternative use of these catalysts requires a fit with the applied polymerization technology to reach the same targeted polymer grades [16].

This interlink between process technology and catalyst structure, the perfect fit between engineering and chemistry, is one of the foundations of the Borstar^®^ PP process. The details of its development and present capability in terms of wide product range and good economics will be discussed in this review. To the best of our knowledge, this is the first attempt at drawing direct connections between process and catalyst development on the one hand and the resulting polymer structure and application performance on the other hand.

## 2. Borstar^®^ PP Technology

Borstar^®^ polypropylene (PP) is Borealis’ proprietary low-pressure, catalytic technology for manufacturing a full package of polypropylene materials. The whole process can be divided in the following areas [17]:Feedstock preparation;Reactor area;Recovery area;Dry end (comprising pelletizing);Material handling, comprising bagging and storage area.

The reactor part of the Borstar^®^ polypropylene modular technology consists of a series of slurry-loop and gas-phase fluidized bed reactors (GPR), where each reactor can be independently controlled. Typically, an additional pre-polymerization loop reactor precedes the reactor cascade. Homopolymers, random copolymers as well as heterophasic matrix copolymers can be produced by employing this technology that enables tailoring the molecular weight distribution (MWD), isotacticity and comonomer content (CC) along the MWD.

The Borstar^®^ PP process technology exhibits a number of competitive advantages, including:Very high catalyst productivity due to high-temperature operation;Very wide product window and independent reactor control, enabling the tailoring of the MWD and CC along the MWD in both the matrix and rubber part of the product;High once-through monomer conversion due to propylene conversion from the loop effluent to the first GPR;Competitive monomer factor and energy consumption;Robust reactor operability and reliability.

According to the Borstar^®^ PP process [18], the first reactor of the series is continuously fed with catalyst, cocatalyst, donor, propylene, comonomer (if desired for the production of random copolymers) and hydrogen. Typically, heterogeneous Ziegler–Natta or single-site catalysts are employed. The polymerization reaction takes place at the active centres that represent the polymerization loci, which are homogeneously dispersed in the catalyst particles. During the first stages of polymerization, the catalyst particles undergo a controlled fragmentation process due to the production of polymer. As polymerization proceeds, the initial catalyst particles remain encapsulated within the gradually growing polymer particle. The polymer particles produced in the first reactor are then continuously fed to the next reactor(s) of the series, where the polymerization reaction continues to take place. The desired mean residence time per reactor defines the production split (weight percentage of production rate per reactor), as implied by the specific product design. Each reactor can be controlled separately and operate under desired reaction conditions in terms of pressure, temperature and reaction species concentrations. This way, each reactor can vary in production rate as well as the molecular properties of the polymer produced. The process is designed in such way that the desired final polymer formulation is deconvoluted in different fractions of unimodal product in terms of MWD and CC along MWD, produced separately in each reactor, fully controlling the polymer microstructure architecture. Furthermore, the fact that the reaction takes place separately in each particle results in homogeneity at the intra-particle level, and each particle consists of a ‘chemical’ blend of the polymer fractions produced in each reactor of the series (see Figure 2, Figure 3 and Figure 4).

### 2.1. Module 1: Catalyst Preparation and Pre-Polymerization

The employed catalyst feed system can handle catalysts either in dry format or suspended in oil. In the case of an oil-suspended catalyst, the catalyst drum is tumbled in multiple directions to achieve the desired homogeneity, and afterwards the catalyst suspension is pushed into the catalyst feed vessel. Typically, the catalyst slurry comprises a hydrocarbon (mineral) oil and the solid catalyst. The slurry is maintained in a homogeneous state in the catalyst feed vessel at a controlled, constant temperature within the range of −30 °C to 80 °C (preferably between 0 °C and 60 °C) and slightly pressurized by an inert gas (e.g., nitrogen or argon) above atmospheric pressure, while a part of that catalyst slurry is fed into the pre-polymerization reactor. The typical concentration of the solid catalyst particles is in the range of 50 to 500 kg/m^3^ in the slurry. The viscosity of the oil needs to be in the range of 20 to 3000 mPa s to prevent settling of the catalyst and to ensure that the catalyst is fed into the reactor with a high level of accuracy and consistency; normally, hydrocarbon grease is added for this purpose. Downstream, at the metering pump, the catalyst slurry is diluted with a propylene flow to increase velocity and assure the smooth flowability of the catalyst slurry to the reactor. Optionally, the catalyst slurry is mixed with an activator and/or an electron donor upstream of the pre-polymerization reactor depending on the selected pre-activation catalyst procedure. Additional components, such as antistatic agents and drag-force-reducing agents, can be added to further improve operability and efficiency [18].

The catalyst is then fed into the pre-polymerization loop reactor, acting as a preconditioning reactor under milder conditions (e.g., the typical operating temperature ranges from 10 °C to 45 °C). This process step ensures controlled catalyst fragmentation, resulting in smooth operation in the upcoming reaction stages [19].

Typically, only a small amount of polymer is produced in this reactor, with the polymer-to-catalyst ratio ranging from 10 to 1000 g PP/g catalyst. Ethylene and hydrogen may be optionally fed to the reactor. A special advantage can be obtained by adding small amounts of ethylene in the range of 0.2 to 1.5 mol%, improving the morphology of the pre-polymerized polypropylene particles [20].

### 2.2. Module 1: Loop Reactor and First GPR

The loop reactor (see Figure 2) is operated at a temperature range of 60–95 °C. Fresh propylene is fed into the loop reactor together with hydrogen to control the molecular weight, and ethylene is fed into the reactor in the case of copolymer production. The residence time in the loop reactor is typically less than one hour. Homo-polymers or random copolymers can be produced in the loop reactor. When producing a copolymer, the comonomer content typically ranges between 0.1 and 8 mol%. Most copolymers are produced using ethylene as a comonomer. The melt flow rate for a 2.16 kg load (MFR2) of a copolymer may vary from 0.1 to 500 dg/min. The operating pressure needs to be sufficiently high in the loop reactor (typically ranging from 30 to 70 bar) to establish a single continuous phase of liquid propylene and avoid the presence of a separate gas phase. In the unfortunate event that this is the case, cavitation effects could potentially result in poor mixing conditions and flowability issues as well as mechanical damage to the loop pump and the mixer.

The loop reactor typically has multiple outlets. Special advantages are obtained when the first outlet is located at a location in the loop reactor where the polymer solids concentration is higher than or similar to the average solids concentration in the reactor and when a second outlet is placed at a location in the loop reactor where the solids concentration is lower than the average polymer solids concentration in the reactor [21].

The loop reactor of Borstar^®^ PP is designed to be operated in supercritical conditions. This feature further extends the operating window for hydrogen and ethylene because gas bubble formation, which is a limiting factor, is avoided, while increasing at the same time their solubility in the polymer phase. Moreover, at supercritical conditions the solubility of the polymer in the liquid propylene decreases, thus resulting in less risk of reactor fouling due to fouling on the loop reactor walls.

The first GPR (see Figure 2) is a fluidized bed reactor, and the polymerization takes place in a fluidized bed formed by the growing polymer particles in an upwards-moving gas stream. The GPR consists of a cylindrical part named the fluidization bed, where gas–solid fluidization takes place above a distributor plate, and the disengagement zone used to prevent the entrainment of polymer particles in the outgoing circulation gas stream. The effluent from the loop reactor is directly fed into the GPR, while the liquid propylene is vaporized and consumed in the first GPR [22]. This enhances the heat-removal capability of the GPR and improves the propylene conversion per pass, resulting in a lower amount of propylene that needs to be recycled. In the GPR the polymer particles, still containing the active catalyst, come into contact with the reaction gases, such as propylene, comonomer(s) (i.e., ethylene, 1-butene, 1-hexene) and hydrogen, which cause polymer to be produced within the particles.

The fluidized bed reactor is equipped with a specially designed gas distribution plate in order to minimize stagnant zones or areas of poor mixing where polymerizing particles could form sheets or chunks [23]. The distribution plate enables part of the fluidization gas to flow along the inside of the reactor wall in the area where the distribution plate normally adjoins the reactor wall. The polymer particles coming from the loop reactor are fed into the GPR via a feed pipe that is usually connected to the GPR at a point above the distribution plate at a position higher than one-third of the effective diameter of the distribution plate.

The upwards-moving fluidization gas stream is established by withdrawing a circulation gas stream from the top of the reactor, typically at the highest location. The gas stream withdrawn from the reactor is then compressed, cooled and re-introduced to the bottom zone of the reactor. Additional propylene and hydrogen are suitably introduced into the circulation gas line. It is preferred to analyse the composition of the circulation gas, for instance, by using online gas chromatography and adjusting the addition of the gas components so that their contents are maintained at desired levels.

The reactor effluent, comprising polymer particles and the gas mixture from the GPR, is withdrawn continuously or intermittently from the fluidized bed and fed into a vessel (outlet vessel). Part of the gas mixture is separated in this vessel and returned to the first GPR system to a place where the pressure is lower than the pressure of the separation vessel [24]. The settled polymer is transferred to reactor module 2 or to the degassing separator. When transferring to reactor module 2, a second gas mixture might be introduced in the lower part of the vessel or downstream of the vessel to facilitate the pneumatic transport to the second GPR. The vessel is operated essentially at the same pressure as the polymer outlet from the first GPR and at a higher pressure than the second GPR. The second gas flow should be below the minimum fluidization velocity of the polymer particles to maintain a settled moving bed. A counter-current flow pattern, with respect to polymer withdrawn from the first GPR, is established so that the reaction mixture, coming from the first GPR, is replaced.

### 2.3. Module 2: Additional GPRs

Reactor module 2 comprises a second GPR and optionally a third GPR in series (see Figure 3). Reactor module 2 enables the production of impact copolymers. Depending on the required weight fraction of the impact copolymer, one or two additional reactors may be employed. The second GPR is typically smaller in volume and operated at a slightly lower pressure (i.e., operating pressure of 1–2 bar lower) compared to the first GPR. This reactor is responsible for the production of rubber content up to a split level of 25%. The operating temperature ranges from 75 to 90 °C for both the second and the potential third GPR. The gas composition is controlled independently from the previous reactor due to the transfer system as well as due to the special design of the product receivers. Typically, the reaction mixture composition differs for at least one component compared to the first GPR mixture (referring to hydrogen and/or ethylene). When a third GPR is employed, the operating pressure of this reactor is slightly lower than that of the second GPR.

The polymer transfer from the second to the third GPR can be similar to the polymer transfer from module 1 to module 2. More specifically, the polymer material produced in each GPR is withdrawn from a suitable area, preferably from the middle zone of the GPR, via the feed pipe into the outlet vessel, through the top part of the outlet vessel. The polymer material is usually discharged in the form of polymer powder, which can additionally comprise agglomerates. The polymer material is withdrawn from the GPR continuously, having a velocity in the range of 5–15 m/s. Flush gas may be used to enhance the transport of the polymer material from the GPR to the outlet vessel. The outlet vessel has a main, a bottom and a top part. As a matter of definition, the main part has the highest effective diameter, whereas the bottom part has a lower effective diameter than the main part. The top part is merely a closure of the outlet vessel.

The challenge in operating the product outlet vessel is removing the produced polymer powder with the minimum amount of entrained gas mixture coming from the GPR via the vessel outlet. An increased amount of gas(es) in the polymer powder results in waste flaring and can further cause quality problems in the resulting polymer powder. By designing in detail the outlet vessel, especially regarding the selection of the position of the powder surface and of the injection point of the second gas mixture—the barrier gas—on the outlet vessel, the amount of entrained gas mixture in the polymer powder can be significantly reduced [25]. It has to be pointed out that the barrier gas should not disturb the operation of the gas–solids olefin polymerization reactor.

### 2.4. Downstream Area

The PP product is withdrawn from the gas-phase reactor and fed into a low-pressure gas–solid separator (see Figure 4). The vapor is compressed and fed into the recovery area while the solid PP particles are gravimetrically fed into the purge vessel. In the purge vessel, the remaining active catalyst is deactivated by steam, while monomers are stripped of PP particles by nitrogen addition [17]. Propylene and nitrogen can be recovered from the purge gas by a recovery unit utilizing gas-permeable membranes [26]. The polymer is then conveyed to the extrusion and pelletizing area.

Pressurized vapor is fed into distillation columns to recover and recycle propylene, ethylene and hydrogen back to the polymerization reactors. Light components (e.g., inert gases and impurities), heavy hydrocarbon components (e.g., waxes and oligomers) and co-catalyst residues are removed by the recovery area [17].

### 2.5. Process Control

Polymerization reactor control is a key component for achieving stable operation at an industrial scale. The main challenge is to maintain high product-quality consistency while maximizing the production rate and minimizing the grade transition times. Borealis was the first company in the polymer industry to implement a nonlinear model predictive control (NMPC) methodology for reactor control. The proprietary Borealis NMPC advanced process control technology (BorAPC, OnSpot) has been continuously developed and optimized since 1994, when it was initially introduced [27]. BorAPC allows Borealis to reach the full potential of the Borstar^®^ PP technology by offering the possibility of operating closer to the process limits (Figure 5).

Nowadays, all Borstar^®^ PP plants operate NMPC in a closed loop, meaning that the controller is able to decide on the control actions without active input from the operator. The controlled variables (CVs) are the production rate, solids content, melt flow index (correlating to the molecular weight of the polymer) and comonomer content. Alternatively, the hydrogen/propylene ratio can substitute for the melt flow index (depending on the available online equipment of the plant). A separate NMPC controller is applied for each reactor; however, manipulated variable values are always shared downstream, enabling feed-forward control actions. NMPC technology not only ensures optimized reactor operation but further simplifies plant control because the operator only needs to provide the set-point values for the CVs. OnSpot consists of an estimator, a process model and a control algorithm. The model parameters are continuously adapted based on process measurements to account for unmeasured disturbances. The control algorithm minimizes the deviation of the predicted values from the set-point trajectory. The mathematical, non-linear process model is based on first principles (mass and energy balances), which allows OnSpot to control the reactor at all operating points using a single model. The controller sample time is typically 1 min, with a prediction horizon of 3–8 h [28].

### 2.6. Dry End, Quality Control and Sustainability

The Borstar^®^ PP process is supported by highly competent quality control (QC) and online polymer analysis (OLPA) teams, which are responsible for a wide range of polymer characterization techniques in an off-line and online format, respectively. Sampling points, located in appropriate positions in the reactor modules downstream of the deactivation and dry end of the Borstar^®^ PP process (e.g., sampling each reactor separately, upstream and downstream of extrusion), offer the possibility of representative product collection.

The QC lab is fully equipped with state-of-the-art characterization techniques, depending on the plant-specific product portfolio needs. In strong connection with Borealis Innovation Centres, the QC labs are able to reveal, among others, detailed rheological, mechanical, optical, molecular and morphological polymer properties. Typical measurement times (from sampling until the available characterization result is uploaded to the plant database) range from 1–3 h.

When required, the Borstar^®^ PP process can be further equipped with online characterization capabilities, reducing the measurement time to less than 30 min. The OLPA lab is responsible for automated sample collection and characterization, using a variety of potential methodologies, including online NMR (nuclear magnetic resonance), rheometry, gel analysis and pellet analysis (shape and size distribution) techniques.

The dry end comprises powder transport and buffering, extrusion (including homogenization, additivation and pelletizing), pellet transport and final lot homogenization. Twin screw extruders are used to extrude and homogenize the product. The final melt flow can be adjusted by controlled rheology and using a dedicated OLPA rheometer. The pelletizers are underwater granulators where the molten polymer strand is cut and cooled rapidly, forming pellets. The pellets are transported by a water stream into a dryer, and downstream from the dryer, the pellets are classified by size. The pelletized product is conveyed from the surge tank to the blenders, where the lot is blended and classified. Finally, the product is transferred to the material handling area.

The material handling area is typically considered outside battery limits. The product can be conveyed from the blenders to storage silos for bulk transport or to packaging lines to bag or box the product. An elutriator can be installed upstream of the bulk silos and upstream of the packaging line.

The Borstar^®^ PP process can be further equipped with technologies to minimize the emission of hydrocarbons to the atmosphere both from the polymer product and during the process of pellet blending. Extruders can be equipped with vacuum stripping and condensate injection upstream of the vent port. Moreover, silo aeration can be applied downstream of the extrusion. Air from silo aeration and off-gas from propylene and nitrogen recovery membranes are lean in hydrocarbons. These hydrocarbons are then processed by a catalytic oxidation or regenerative thermal oxidation (RTO) unit to reduce hydrocarbon emissions to the atmosphere [26].

## 3. Borealis Catalyst Design

The development of Borealis’ proprietary catalyst started in the late 1980s under the scope of finding an alternative magnesium halide-based support system allowing better design of highly porous catalysts. In combination with a specific internal donor, this should allow the development of a fourth-generation ZN catalyst [8,9] suitable both for the production of high-isotacticity iPP homopolymers and high-impact copolymers with significant amounts of amorphous ethylene–propylene copolymer (EPC).

The standard support type was then micronized MgCl_2_ produced by ball-milling, applying co-milling with TiCl_4_ as a titanization process. The role of the internal donor in generating and stabilizing the active titanium centres and allowing the efficient use of external donors was becoming clear already [29,30], and phthalic acid esters were established as suitable donor compounds. Also, the importance of selecting the right external donor from the family of alkoxysilanes was supported by respective studies [31] finding a combined effect on stereoregularity, activity and molecular weight. All of these detail studies also showed the multi-site nature of such catalyst systems [32].

Several ideas were considered for developing the new catalyst type, with spray-crystallization of a molten magnesium dichloride complex to form the carrier being one of them, for which also a complex of MgCl_2_-hexahydrate with NH_4_Cl was envisioned [33]. Transesterified ZN catalyst types are produced in several stages, starting with the formation of an adduct between MgCl_2_ and an aliphatic alcohol, like ethanol, to obtain MgCl_2_ * 3 EtOH, as already suggested by Chien et al. [30]. This can be spray-crystallized from a melt of ~100 °C into particles of 10–300 µm with atomizing nozzles in a nitrogen atmosphere [34], those being suitable for titanization with an excess of TiCl_4_ in a hydrocarbon solvent (this and all following reaction equations are not to be seen as stoichiometric):MgCl_2_ × n R1OH + (n + m) TiCl_4_ → MgCl_2_ × n TiCl_3_OR1 + n HCl + m TiCl_4_(1)
with n being 1–6 and R1 being, for example, C_2_H_5_. Adding an ester or a carboxylic acid, preferably an alkyl ester or phthalic acid ester like di(isobutyl phthalate), as an internal donor to this titanized carrier, an adduct of all components is created:MgCl_2_ × n TiCl_3_OR1 + n R3COOR2 → MgCl_2_ × n TiCl_3_OR1 × n R3COOR2(2)

This adduct can be transesterified at a temperature higher than 136 °C, i.e., above the boiling point of TiCl_4_, in which process the ester groups R1 and R2 exchange places:MgCl_2_ × n TiCl_3_OR1 × n R3COOR2 → MgCl_2_ × n TiCl_3_OR2 × n R3COOR1(3)

The temperature, solvent choice and duration have been found to affect multiple parameters of the resulting catalyst, like the Ti content, polymerization activity and isotacticity of the resulting PP [35]. Finally, undesired residue materials such as titanium chloro-alkoxides are removed by extraction or washing with a hydrocarbon solvent, receiving an adduct of the carrier and the ester donor coupled to the active Ti centre:MgCl_2_ × n TiCl_3_OR2 × n R3COOR1 × m TiCl_4_ →MgCl_2_ × n R3COOR1 × m TiCl_4_ + n TiCl_3_OR2(4)

For polymerization, this procatalyst is combined with an aluminium-alkyl or -alkylhalide and an external donor as usual. By optimizing the components and preparation steps, catalysts of high porosity, giving the required isotacticity and rather large polymer particles without a too-high “fines” fraction, could be developed [36,37].

This catalyst type has been both studied systematically and employed in multiple polymerization setups. The work by Garoff et al. [38] highlights the extent to which isotacticity and molecular weight distribution can be modified by varying the ratio between the external donor feed and Ti content in the catalyst feed, as shown in Figure 6 for dicyclopentyl dimethoxysilane (donor D). Here, triethylaluminium (TEAL) at a molar Al/Ti ratio of 250 was employed, resulting in a parallel change of the Al/donor ratio. In practice, the additional effects on catalyst activity—having a maximum at a donor/Ti ratio of ~10 in the present case—and the hydrogen response with respect to the molecular weight of the polymer need to be considered. In any case, the well-known correlation between isotacticity and modulus [39] made the transesterified catalysts suitable for producing high-stiffness grades, and the improved stability at higher polymerization temperatures fit well with the Borstar PP technology.

This first generation of Borealis catalysts, commonly designated as BCF types, was subsequently used in the development of complex grades like multimodal random copolymers and high-impact heterophasic copolymers. In a comparative study by the group of McKenna [40], it was found to be suitable for incorporating high amounts of EPC while maintaining full powder flowability, proving the porous nature of the matrix polymer.

Another basic idea for generating an even particle size in catalyst production was the origin of the second generation of proprietary catalysts for the Borstar process. The new emulsion-based catalyst preparation process—commonly called ‘Sirius’—uses liquid precursors in a three-step process [41,42,43], starting with a reaction of butyl octyl magnesium (BOMAG) with 2-ethyl-hexanol to form an Mg-alkoxide:C_4_H_9_MgC_8_H_17_ + 2 C_6_H_12_(C_2_H_5_)OH → Mg(OC_6_H_12_(C_2_H_5_))_2_ + C_4_H_10_ + C_8_H_18_
(5)

This alkoxide, used in excess, is partially allowed to react with phthaloyl dichloride (PDC) to form a complex between magnesium chloride and di(ethylhexyl)phthalate (DEHP), acting as internal donor in the catalyst precursor for the later polymerization: Mg(OC_6_H_12_(C_2_H_5_))_2_ + PDC → MgCl_2_ + DEHP(6)

Reacting this complex with TiCl_4_ results in the formation of a two-phase liquid–liquid system, as shown on the left-hand side of Figure 7. Adding a specific polymeric surfactant allows the formation of an emulsion with droplets in the range of 10–100 µm diameter, which can be converted into a dispersion with solid particles by a temperature increase (details of the reaction are described by Rönkkö et al. [41], with the related conditions in the patents [42,43]). Washing and drying results in perfectly spherical particles which have a smooth surface and no apparent porosity, as shown on the right-hand side of Figure 7.

These catalysts were tested in polymerization both alone [44,45] and in direct comparison to the earlier generation of transesterified types [46,47]. The first question to be answered was whether an apparently non-porous catalyst could yield high productivity at all, and how fragmentation in polymerization would occur. Most previous polymerization models, like the one presented by Cecchin et al. [48], assume fragmentation around primary MgCl_2_ micro-crystals. No such structure is obvious in the emulsion-based catalyst particles, but both an X-ray diffraction analysis and the fragment analysis discussed below indicate its presence. For emulsion-based or self-supported catalysts, different but similar fragmentation processes have been observed, however without significant production of ‘fines’ by excessive growth stresses. The rather narrow particle-size distribution and spherical shape of the original catalyst particles are retained very nicely, analogous to the ‘replica effect’ for other spherical catalyst types [44]. After a single- or two-stage polymerization of a homopolymer or random copolymer, MgCl_2_ fragments in the size range below 100 nm were observed [45,49].

This first generation of emulsion-type catalysts has, like the transesterified catalyst based on spray-crystallized support, a phthalate-type internal donor formed in situ from the precursor pthaloyl dichloride, but the resulting polymers differ significantly. For homopolymers, a narrower MWD and lower oligomer content but also a greater flexibility in terms of isotacticity control by the external donor feed were observed [46], enabling high-flow grades for fibre and moulding applications with low emissions. Even more relevant are the differences for ethylene–propylene (C2C3) random copolymers, where a higher fraction of isolated C2 units—commonly called ‘improved randomness’—was found [47]. This results in a lower melting point at a given comonomer content and results in an improved sterilization resistance [50]. In line with this performance, this catalyst type was also found suitable for the production of soft random-heterophasic copolymers (RAHECOs), the details of which will be discussed in the following chapter.

Limitations in the incorporation of high EPC amounts led to the development of an emulsion-based catalyst type with dispersed micro- or nanoparticles like Al_2_O_3_ or SiO_2_ [43,51,52]. These particles, while inactive for polymerization, are capable of generating additional cavities in the growing polymer particles during the early stages of polymerization (matrix stage), allowing a more evenly distributed EPC dispersion in further stages [53].

The ‘Sirius’ technology was also adopted for developing Borealis’ own line of post-phthalate ZN catalysts, although first attempts in that direction had been conducted on a more conventional type of support [54]. This general trend of applying alternative internal donors instead of phthalates was motivated largely by health and environmental concerns over this substance group. Certain phthalates are known to have estrogenic activity [55], but the respective discussion was rather incited by the high amounts used as plasticizers in poly(vinyl chloride) (PVC) than by the minute amounts in polyolefin catalysts. The residual content of internal donors or their decomposition products in iPP is, because of the high catalyst activity, in the range of less than 0.1 ppm.

Post-phthalate ZN catalyst development was started by LyondellBasell with diethers and succinates [13,48,56,57] as internal donors. The respective catalysts developed at Borealis use a citraconate as an internal donor [58] and have been found to be suitable for the commercial production of iPP homopolymers [59] and copolymers with ethylene [60]. The change in internal donor again changes the comonomer incorporation, as observed for the earlier generations by Vestberg et al. [61], resulting in a lower melting point at a given ethylene concentration (see Figure 8).

This catalyst type is also employed for producing heterophasic ethylene–propylene copolymers (HECOs) with high impact strength in a wide range of EPC content, achieving free-flowing polymer powders up to an XCS content of 37 wt% [16]. As in random copolymers, reactivity towards C2 is higher than for ZNC types with phthalate donors, causing differences in the C2 content of the XCS fraction and Tg(EPC). Further developments broadening the performance range of copolymers can be expected.

A special catalyst-related technology also developed at Borealis is pre-polymerization with a suitable monomer to produce a high-melting-point polymer acting as a nucleating agent for iPP. Due to catalyst fragmentation, this process allows a dispersion of pre-polymer particles in the final polymer, giving massive nucleation effects at a very low concentration [62]. One of the best candidates for in-reactor nucleation is isotactic poly(vinyl cyclohexane), PVCH, for which a high degree of lattice-matching to several planes of the α-phase of iPP has been found by the group of Lotz [63]. Isotactic PVCH is highly crystalline with a melting point of ~360 °C and can be produced by polymerization with Ziegler-type catalysts, as already described in the 1960s [64]. The interest in this period was, however, more on the atactic counterpart produced by the hydrogenation of polystyrene and seen as a possible competitor to polycarbonate [64,65]. A patent application for its use as nucleating agent for iPP was filed by Sumitomo in 1983 [66], already highlighting the high activity with an increase in crystallization temperature (Tc) of more than 10 °C at 29 ppm of PVCH at low concentrations. The polymer was marketed by Sumitomo as masterbatch for nucleation under the trade name CAP, with Kakugo highlighting its performance in several papers and actually claiming nucleating activity at 1 ppm already [67].

This process was optimized by Borealis in several respects, including the use of a transesterified ZN catalyst as described above in combination with a silane-type external donor for optimizing the PVCH structure and an elevated temperature for shortening the pre-polymerization time [68]. Combining this with a second pre-polymerization step with propene and optimization of the isotacticity, stiffness levels of more than 2 GPa could be reached for homopolymers, and the process was also expanded to random and heterophasic copolymers [69]. Structurally similar but alternative polymeric nucleating agents like poly(cyclopentene), or PCP, were also found to work at very low concentrations [70] but did not succeed technically (Lee and Yoon also compared the effects of PVCH and PCP to other polymeric nucleating agents like PA-66 and PPEK in their study).

More recently, this family of efficient polymeric nucleating agents has been broadened further by poly(trimethylallyl silane), PTMAS [71], and poly(vinyl cyclopentane), PVCP [72]. While all of these are efficient nucleating agents for iPP and also can prevent the formation of the mesomorphic phase at higher cooling rates (i.e., in quenched samples), the efficiency under the latter conditions is still best for PVCH, as shown in Figure 9, where the doubling of the points at higher cooling rates indicates the combined formation of the α- and mesomorphic phase. This makes the technology especially suitable for high-isotacticity film grades combining high modulus and low water vapour permeability even under cast-film conditions [73], offering an economical alternative to biaxially oriented or specially coated films in the packaging of moisture-sensitive products (an example for such a grade is HD905CF of Borealis). It works similarly in heterophasic copolymers for applications requiring high impact strength at sub-zero temperatures, outperforming other nucleating agents and ensuring stable α-phase formation up to more than 1000 K/s [74]. High-flow grades designed in this way are particularly suitable for thin-wall injection moulding applications, as in the case of BJ368MO. Here, the higher cooling rate still allowing stable crystallization results in a significant cycle-time reduction.

The in-reactor-produced iPP/PVCH combination can be varied in concentration [75] and is capable of interacting positively with reinforcing fillers like talc or glass fibres. In this way, PVCH is also suitable to compensate for the unwanted nucleating effects of certain pigments. Moreover, the process has also been adapted to work for post-phthalate ZN catalysts based on an emulsion process, as also detailed above [76]. As a non-migrating nucleating agent of high thermal stability, it is perfectly suited for high purity requirements, a frequent request in the post-phthalate era of iPP design.

## 4. Borstar PP Composition and Performance Range

One of the main advantages of using the combination of a liquid bulk loop reactor and a fluidized gas-phase reactor (GPR) in series for producing homopolymers and random copolymers with C2 or higher α-olefins like 1-butene (C4) or 1-hexene (C6) is the higher difference in average molecular weight and comonomer content values that can be achieved between the two fractions [77,78]. The use of bi- or multimodality for broadening the MWD with the target of increasing flow-induced crystallization and stiffness [79] is based on the role of longer molecules in the formation of the stable nuclei and shish-kebab structures mostly studied in solution or extruder blends [80] and found to increase modulus significantly [81]. This approach is, however, not practical at a large scale; reactor-based bimodality is a suitable alternative which has been applied successfully in polyethylene design [82].

Bimodal homopolymers with a significant MFR difference between the loop reactor and GPR were among the first innovative products from the Borstar^®^ PP process (the importance of excellent homogenization for utilizing the enhanced rheological performance has just been highlighted recently again [78]). Combining increased polydispersity with high isotacticity and selective nucleation gave advantages not only in stiffness, but also in properties otherwise requiring massive orientation in processing like the water vapour impermeability of films [73]. Shortly afterward, this design principle was also applied to produce advanced random copolymers [83,84,85] in the loop/GPR design. PP homopolymers and C2C3 random copolymers are generally miscible, as long as the comonomer content does not exceed ~8 wt.% for most ZN catalyst types, and as long as the molecular weight difference between the two components is not too high [78,86], resulting in a single-phase structure allowing high transparency and low haze.

For film applications, this allows a broader melting range, resulting from a wider distribution of lamellar thickness and leading to a broader sealing window, i.e., a lower sealing initiation temperature (SIT) and/or a higher sealing end temperature (SET) [84]. Likewise, the sterilization behaviour and especially the loss of transparency in steam sterilization can be influenced by having a low comonomer content fraction in the composition [85]. Another relevant application area is thin-wall injection moulding requiring an MFR level of 70–100 g/10 min, for which Figure 10 presents a direct comparison of mono- and bimodal C2C3 random copolymers. In food packaging, for example, the design of cups and closures requires materials with a good balance between haze and impact strength, but also heat resistance and solubles content. The bimodal materials presented here all have a total C2 content of ~4 wt.%, but with a significant split between the loop and GPR fractions, allowing an improved property balance [87], with a typical example being BorPure™ RJ766MO. Combining this design principle with the aforementioned in-reactor nucleation allows for the production of parts with stable long-term performance at low cycle times, i.e., high cooling rates [82]. This design scheme can be expanded even further when applying single-site catalysts with their inherently better comonomer incorporation and lower level of extractables [78,88].

The Borstar^®^ PP Module 2 setup (pre-polymerization, loop and two GPRs in series) is able to further expand the product properties window by introducing a trimodal copolymer design, where both the MWD and CC vary between the three distinct fractions. Compositions for hot-water and pressure pipes are one possible target here [89], as the inherently low MFR for this application allows a wide spread of the molecular weights of the fractions, such as over three orders of magnitude between 0.002 and 2 g/10 min. Trimodal compositions may also include a homopolymer fraction and ultimately a very propylene-(C3)-rich disperse phase (EPC), similar to the concept of ‘interpolymers’ developed for combining transparency and impact strength in the area of sterilizable packaging [90,91] (the Borealis grade Borpact™ BC918CF is an example of such a grade). To demonstrate the resulting phase structures, examples of a HECO with a C3-rich and C2-rich disperse phase are presented in Figure 11. The difference between the homogeneous EPC particles having a diffuse interface to the matrix in the case of (a) and the core-shell particles with a smoother interface to the matrix in the case of (b) is obvious. Transmission-electron micrographs based on RuO4-contrasted specimens are used here, as this technique gives a very high resolution down to the lamellar level. A more complete series of images and the related properties can be found in the work of Grein et al. [90]. Nevertheless, a study of the onset of phase separation in the case of ZN catalysts is still not available in the literature. For single-site catalysts, this is expected within the range of 10 to 22 wt.% of C2 in the xylene cold soluble (XCS) fraction, as also indicated by the appearance of a second glass transition for the EPC [92], but the matrix design and molecular weight are likely to play a role as well in practice.

These rather complex designs, clearly beyond the ‘standard toolbox’ of HECO design [93], bring positive effects in high-flow grades for bottle closure applications. The integration of hinge caps or tethering elements into soft-drink caps is required to avoid littering and microplastic formation, and a trimodal copolymer design provides the necessary combination of stiffness, impact strength and transparency for this application [94].

The availability of a second GPR further allows the production of the highest possible EPC content value, which is necessary for automotive applications to obtain PP grades of optimum stiffness–impact balance [95], resulting in grades like EF015AE of Borealis. More recently, this has also been demonstrated to be possible with post-phthalate catalysts [16]. Maximizing the EPC content may also be combined with bimodal EPC design in order to manage the manifold requirements of the application, simply because different targets may require different components. The already mentioned variation of the ethylene content of the disperse phase, characterized by XCS, is a typical example here [96,97]. A high amount of ethylene in the rubber phase triggers incompatibility and the development of high interfacial tension with the PP matrix. Consequently, the disperse phase tends to reduce the interface area per volume with the matrix, resulting in bigger particles. Moreover, due to the statistics of ZN catalyst-based polymerization, the amount of crystalline EPC—tending in structure towards a linear, low-density PE with C3 as a comonomer—increases likewise, causing enhanced structural diversity inside the EPC particles and additionally affecting the performance [98,99]. While this is largely beneficial for impact strength, also owing to a decrease in the glass transition of the disperse phase, it may cause problems in surface appearance and paint adhesion.

Heterophasic copolymers (HECOs) with a bimodal EPC design in terms of C2-content and/or molecular weight, as expressed by the intrinsic viscosity IV(XCS), can be used advantageously both as pure polymers for moulding applications [100] and in compounds with mineral fillers for automotive applications [101]. Figure 12a shows one possible morphology for such systems. When one of the EPC fractions exhibits sufficiently high molecular weight, the polymer particles become more stable against deformation in flow and aggregation, as observed for monomodal EPC systems [90,96]. Such reactor-based systems should be distinguished from blends using high-density (HDPE) [102] or low-density polyethylene (LDPE) [103] for modifying the overall disperse phase composition, although such compositions are efficient in reducing stress whitening or improving the balance between transparency and impact strength.

Aside from designing bimodal EPCs and/or maximizing EPC-content, a Borstar^®^ four-reactor setup also enables designing HECOs with a trimodal matrix. The matrix delivering the stiffness of such materials is built to contain fractions of high, medium and low molecular weight.

The beneficial effect of the high-molecular-weight part is very likely based on the orientation of the long chains leading to more flow-induced orientation and higher stiffness as well as on an increase in the nucleation density of the material. The low-molecular-weight fraction still allows good processing of these polymers, whereas the medium fraction is needed for compatibilization reasons (see Figure 13a). Applying this polymer design, the stiffness can be further increased, and materials of extraordinary rigidity are obtained (see Figure 13b). Certainly, as in all industrial polymerization processes, the design freedom is also limited for trimodal matrices from a process point of view by the final MFR of the product, with process limitations like flash points between reactors and catalyst activity. Nevertheless, currently available fourth+-generation ZN catalysts combined with a Borstar^®^ PP four-reactor setup allow for the production of HECOs with trimodal matrices for materials suitable for pipe applications to thin-wall injection moulding [104,105,106], like BH381MO of Borealis.

Combining the high-stiffness trimodal matrices with a well-designed EPC, copolymers with an outstanding stiffness–impact balance can be achieved (see Figure 14). Not only is the stiffness superior due to the high-stiffness matrix, but the impact performance is also significantly improved compared to benchmarks of similar EPC content levels. It is suggested the trimodal matrix design fosters—also at relatively high end flowability and therefore an unfavourable viscosity ratio between the disperse and matrix phases (ηEPC/ηmatrix)—a morphology with fine and well dispersed EPC particles due to the presence of the high-molecular-weight fraction.

Apart from the EPC design discussed earlier, the matrix can also be modified, substituting the PP homopolymer with a RACO with C2 or higher α-olefins. This leads to random-heterophasic copolymers (RAHECOs), the property combination of which results from the presence of the random propylene copolymer (RACO) matrix and the dispersed elastomeric phase (EPC). An early example of such grades is the still-existing SD233CF of Borealis.

The properties of a RAHECO can be varied by different parameters, many of which are intrinsically linked to each other. In an early development phase [107], the overall modulus was found to result mostly from the comonomer content of the matrix phase and the EPR content (see Figure 15). Optimizing the toughness and optical performance of such compositions, however, requires adapting the phases’ compositions and their respective molecular weights, reaching a morphology, as shown in Figure 12b. Optimum transparency cannot be achieved with a fully amorphous EPC fraction, as shown in Figure 11a, but rather with a density match between the matrix and disperse phase, which also minimizes the difference in the refractive index and can equally well be reached with post-phthalate ZN catalysts [108].

Fully ductile behaviour at ambient temperature can be achieved at EPC contents of 15–20 wt.% for a pure RACO matrix, which is related to the transition from crazing to shear yielding below a certain EPC particle distance [109]. In detail, however, this transition will also be affected by the matrix stiffness and the molecular weight of the EPC phase [110]. For producing even softer PP-based materials, targeted at replacing LDPE or even plasticized poly(vinyl chloride), a careful look at the powder morphology was found to be necessary. The essential morphology development of HECOs was described already in the work of Cecchin et al. [111] as the gradual filling of the crystalline and porous particles from the matrix stage(s) of the process, and it was later studied by the group of Kosek in more detail both by modelling [112] and by advanced structural characterization methods [113]. Under the conditions of the gas-phase reactor and the following ‘dry end’ part of the polymerization plant, i.e., at temperatures of 70–85 °C, the EPC fraction is highly mobile and potentially also plasticized by the monomer mixture, making a fine dispersion inside the particle without significant surface fractions decisive for maintaining powder flowability and plant operability. A sufficient porosity of the matrix particle and the absence of particle breakup are clearly necessary for this, especially at high EPC contents.

Considering this, the Borstar^®^ PP process with a four-reactor configuration allows the stable production of various ‘Soft-PP’ grades with a bimodal RACO matrix and an EPC content up to 45 wt.% [114]. The mechanical profile of these and earlier-developed RAHECO grades is compared to those of RACOs and HECOs in Figure 16. ‘Soft-PP’ grades for medical applications like infusion pouches or blow–fill–seal containers, combining toughness and transparency with a ‘collapsing’ emptying performance and sufficient stability in steam sterilization, are so far the endpoint of development [115], exemplified by Bormed™ SB815MO.

## 5. Conclusions and Outlook

This review presents modern polypropylene production as an integrated concept between the process technology, catalyst system and polymer design for the example of the Borstar^®^ PP process. Multiple interactions exist between the corners of this ‘performance triangle’, which have been highlighted in the previous chapters, like between comonomer response and melting point range or between EPC capacity and impact strength. The constantly changing economic and general environment, however, requires constant adaptation of such processes, especially regarding the considerations of sustainability and reducing the carbon footprint of production.

At present, an ongoing challenge is the integration of single-site catalyst (SSC) systems into the Borstar^®^ PP process. While this catalyst type clearly offers advantages in terms of narrow MWD and low emissions [116,117], in the incorporation of higher α-olefins like 1-butene or 1-hexene [78,118] and also in the design of advanced multiphase systems [92,119], it also presents new challenges regarding cost and plant operation. The coming years will tell whether SSC-PP can become a success story, as in the case of PE [120].

Ultimately even more important will be the need for incorporating monomers from sustainable and largely carbon-neutral sources like renewable-based [121] and chemical-recycling processes [122,123]. This is not optional for the polymer industry—it is a clear requirement, despite the already rather low carbon footprint of PP in comparison to other polymers. Monomers from such sources present new challenges in terms of purity with respect to contaminant type, which are likely to require adaptations in catalyst systems and monomer purification.

Reducing CO_2_ and other greenhouse gas emissions while generally saving energy in all stages of the process, but already starting at hydrocarbon cracking, is one more aspect of this challenge. Improved integration of energy streams, including external use of cooling water for district heating purposes, but also better process control and extended use of process digitalization [124] are bound to play an elementary role here.

Since its original development, PP has solved a large number of technical and societal problems for humanity. It will be the common task of science and industry to further improve the balance between the beneficial role and the negative side effects of its production.

## Figures and Tables

**Figure 1 polymers-14-04763-f001:**
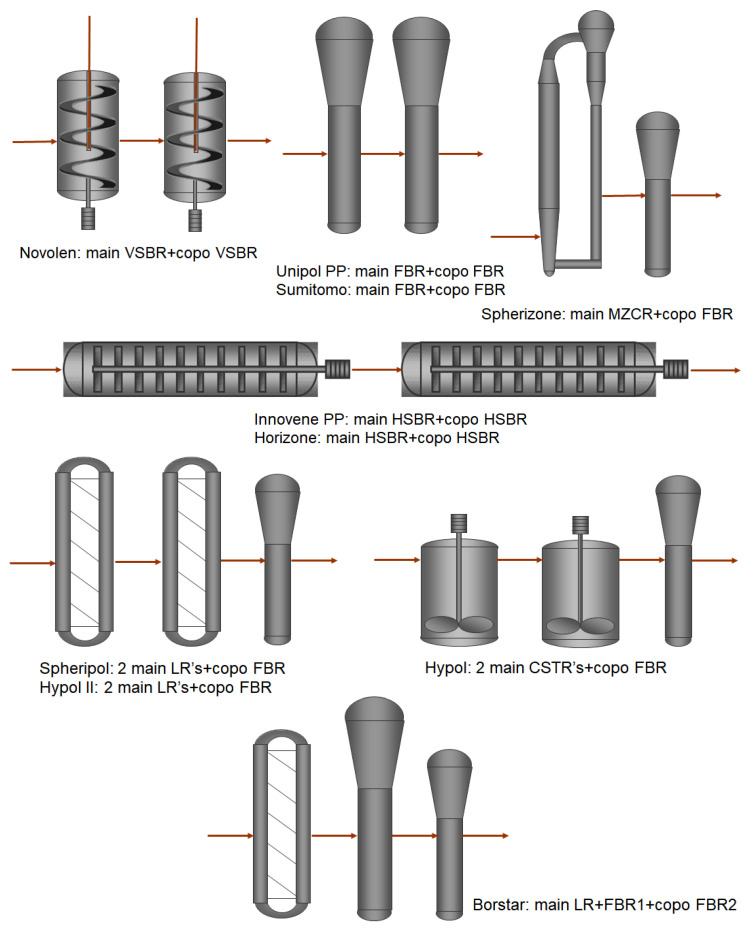
Generalized schemes of the modern polypropylene production processes.

**Figure 2 polymers-14-04763-f002:**
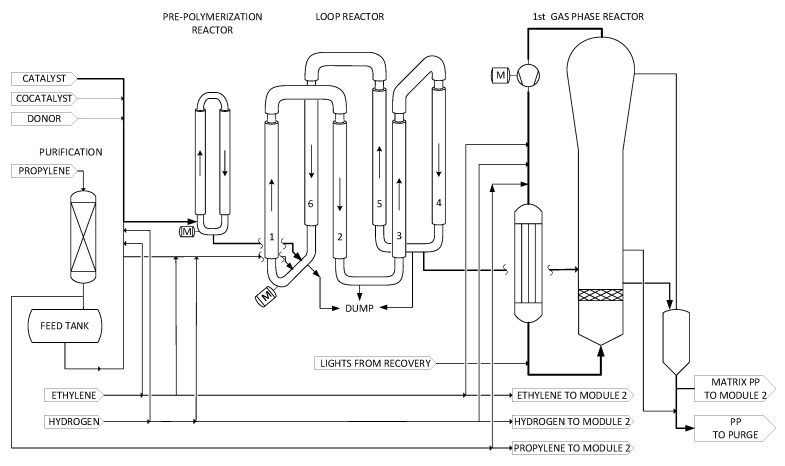
Simplified schematic flow diagram of reactor module 1 and feeding of the Borstar^®^ PP process.

**Figure 3 polymers-14-04763-f003:**
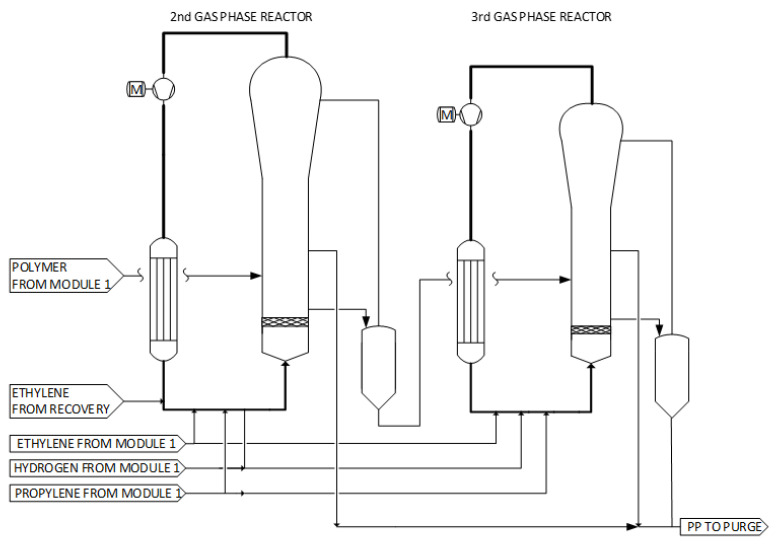
Simplified schematic flow diagram of reactor module 2 of the Borstar^®^ PP process.

**Figure 4 polymers-14-04763-f004:**
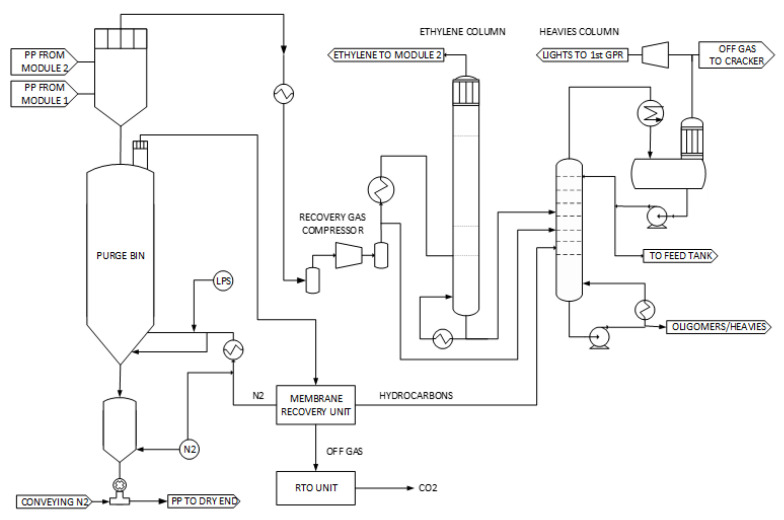
Simplified schematic flow diagram of the purge and recovery section of the Borstar^®^ PP process.

**Figure 5 polymers-14-04763-f005:**
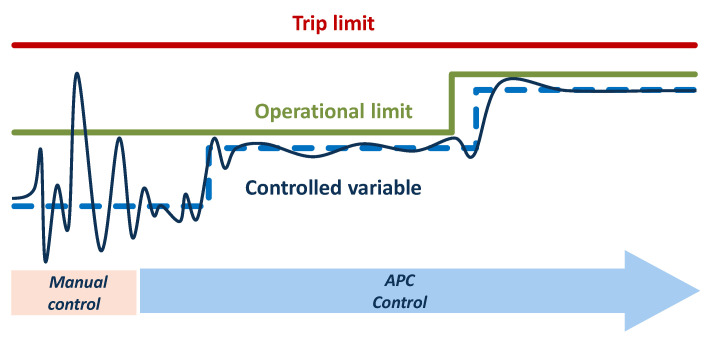
Temperature–time profile for analysis of structure formation at different cooling rates. The final, red-coloured heating segment served for analysis of the fraction of crystals in the prior cooling step.

**Figure 6 polymers-14-04763-f006:**
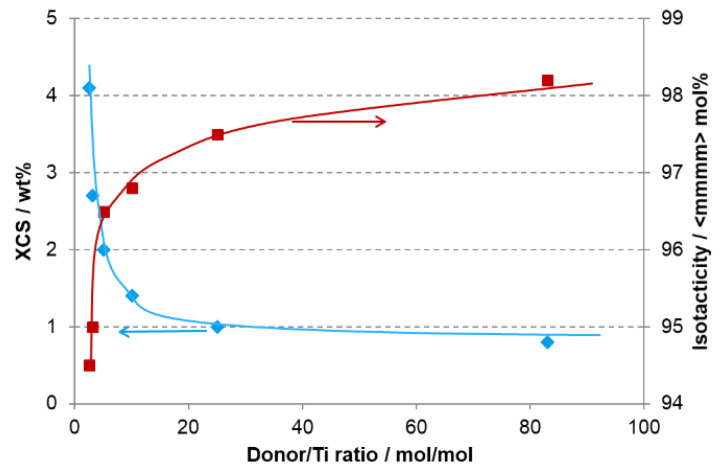
Effect of the molar ratio between donor and Ti on isotacticity (red, pentad regularity by ^13^C-NMR) and xylene cold soluble (blue, XCS) fraction (data from [38]).

**Figure 7 polymers-14-04763-f007:**
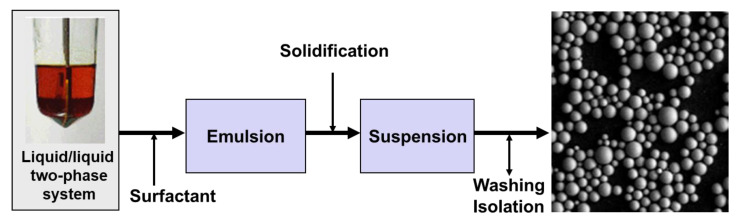
Process scheme of the ‘Sirius’ emulsion process for ZN catalyst production.

**Figure 8 polymers-14-04763-f008:**
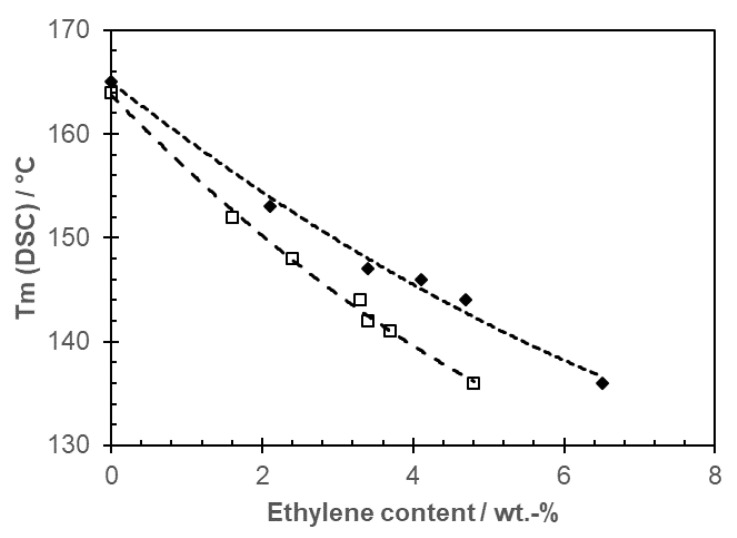
Melting point as function of ethylene content for monomodal random copolymers based on emulsion-type catalyst with phthalate (full symbols) and citraconate (open symbols; data from [47,60]).

**Figure 9 polymers-14-04763-f009:**
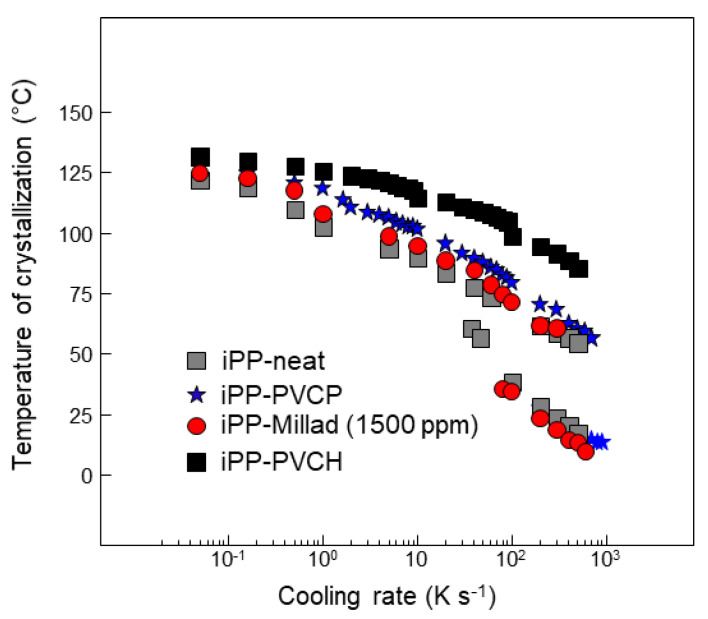
Crystallization temperature (Tc) as a function of cooling rate for a non-nucleated (neat) iPP homopolymer (grey squares), two iPP homopolymers with pre-polymerized nucleating polymers, PVCP (blue stars) and PVCH (black squares), and one conventionally nucleated iPP homopolymer with DMDBS (red circles; data are a combination of results in [72] and previously unpublished data).

**Figure 10 polymers-14-04763-f010:**
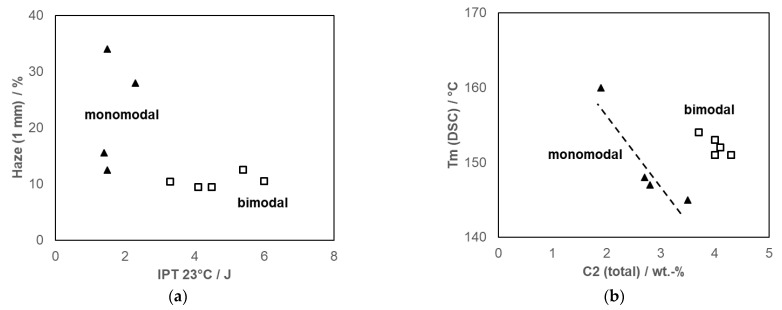
Comparison of high-flow mono- and bimodal C2C3 random copolymers: (**a**) balance between toughness (total energy from instrumented puncture test) and haze, and (**b**) correlation between total C2 content and melting point (data from [87] and commercial references).

**Figure 11 polymers-14-04763-f011:**
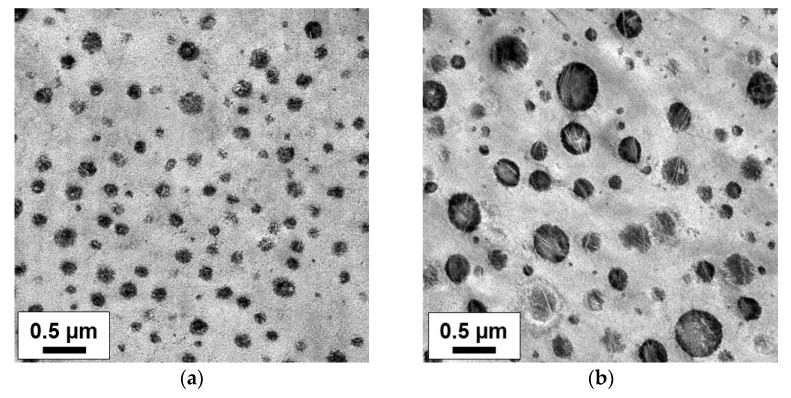
Phase morphology overview of heterophasic copolymers (HECO with homopolymer matrix) from the Borstar^®^ PP process: (**a**) ‘interpolymer’ with C3-rich EPC (12 wt.% XCS with 24 wt.% C2(XCS)) and (**b**) conventional type with C2-rich EPC (12 wt.% XCS with 57 wt.% C2(XCS)).

**Figure 12 polymers-14-04763-f012:**
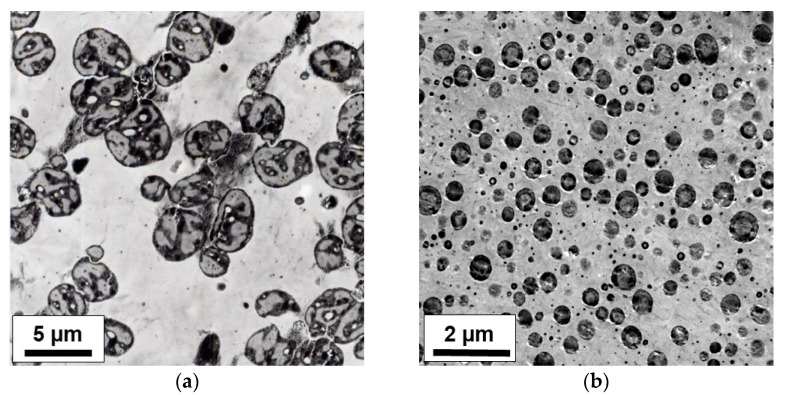
Phase morphology overview of special HECOs from the Borstar^®^ PP process: (**a**) reactor-based thermoplastic polyolefin (RTPO) with bimodal EPC having significant crystalline fraction (XCS: 32 wt.-%) and (**b**) RAHECO with random propylene copolymer matrix (4 wt.-% C2) and 15 wt.% EPC (total XCS: 22 wt.-%).

**Figure 13 polymers-14-04763-f013:**
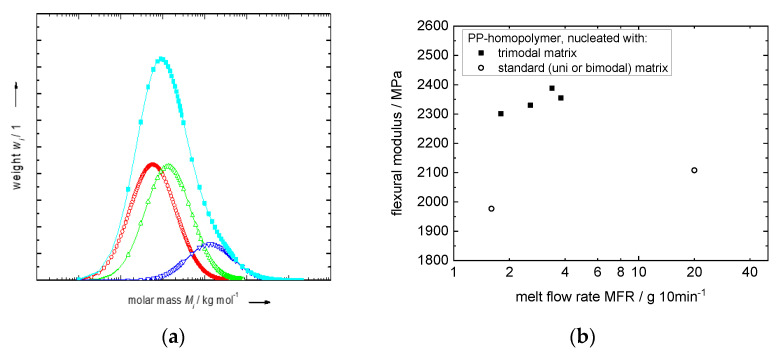
(**a**) Schematic representation of the MWD of a trimodal PP homopolymer (curves with different colour represent the different fractions) and (**b**) flexural modulus over MFR for selected tri-modal and conventional PP homopolymers of medium flowability (data from [105]).

**Figure 14 polymers-14-04763-f014:**
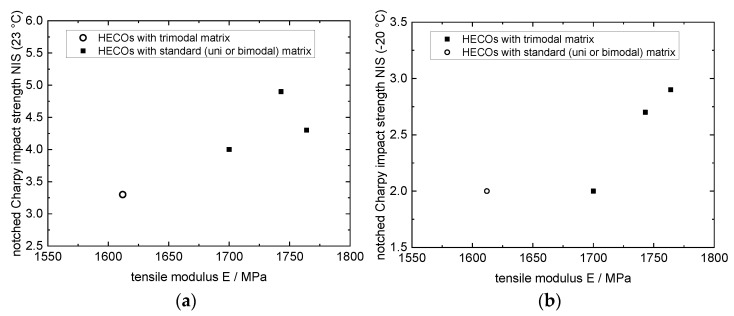
Stiffness–impact balance of HECOs with trimodal matrix compared to commercial references with unimodal matrix with NIS measured at different temperatures: (**a**) 23 °C and (**b**) −20 °C (data from [106] and commercial grades).

**Figure 15 polymers-14-04763-f015:**
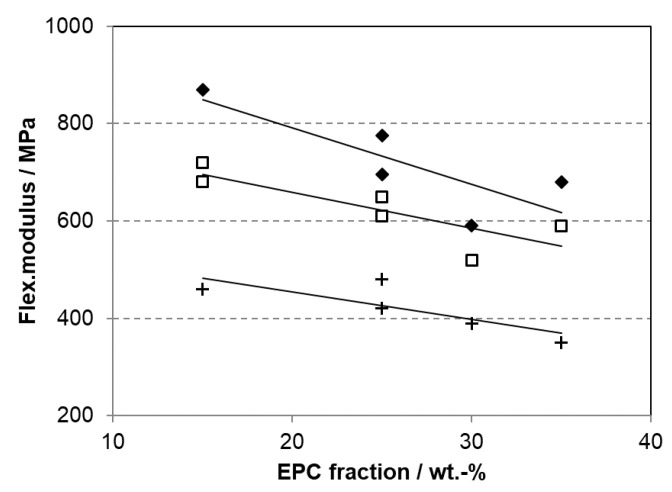
Flexural modulus of random-heterophasic ethylene–propylene copolymers (RAHECOs) with different C2 contents of the matrix (♢ 4 wt.%, □ 6 wt.%, + 8 wt.%) and EPC content (C2 of EPC ~36 wt.% C2, data from [107]).

**Figure 16 polymers-14-04763-f016:**
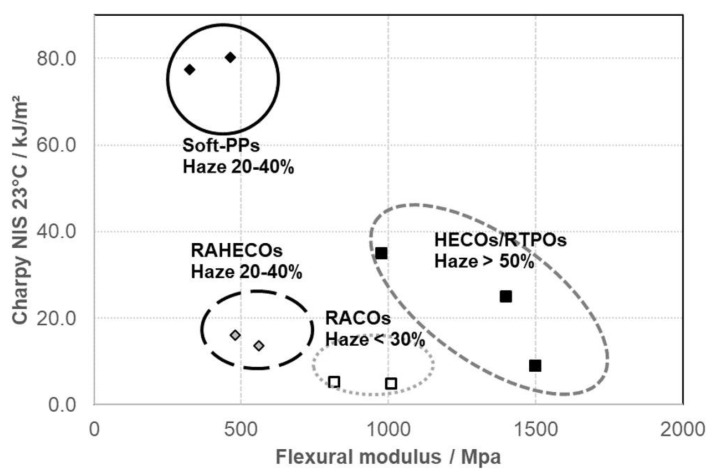
Stiffness/impact balance of different standard and advanced PP copolymers (points correspond to commercial or developmental grades, haze range on 1 mm injection-moulded plaques indicated for each material class).

## Data Availability

Further data can be made available by the corresponding author upon personal request.

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
