# Peer review of "Polymerization in the Borstar Polypropylene Hybrid Process: Combining Technology and Catalyst for Optimized Product Performance"

_polymers, 2022, doi:10.3390/polym14214763_

Round 1
Reviewer 1 Report
Dear Authors,
This review describes the Borstar PP technology that is developed to improve the performance of polypropylene synthesis by combining catalysts, process technology, and polymer design.
The manuscript is well-written and organized. It has covered all the other previously developed technologies along with their merits and demerits.
I recommend accepting this manuscript with no further changes.
Thank you
Author Response
We thank the reviewer for his entirely positive view on our paper.

Reviewer 2 Report
The authors provide a good overview of the history of polypropylene polymerization development, especially of bimodal polypropylene polymerization. The authors' language is fluent and clear, and the reviewers believe that the authors are ready for publication after making certain changes.
(1) The authors refer to a large number of Borstar® polymer technologies in the paper and hope to add some typical grades so that readers (especially engineers) can understand the differences between the different grades of polypropylene.
(2) The authors could add some rheological data, especially the difference between single-distributed PP and bimodal cloth PP in shear rheology and capillary rheology tests.
(3) The authors should add some crystallization kinetic data such as activation energy, semi-crystallization time and other parameters to the section on crystallization properties to make a comprehensive addition.
The authors provide a good overview of the history of polypropylene polymerization development, especially of bimodal polypropylene polymerization. The authors' language is fluent and clear, and the reviewers believe that the authors are ready for publication after making certain changes.
(1) The authors refer to a large number of Borstar® polymer technologies in the paper and hope to add some typical grades so that readers (especially engineers) can understand the differences between the different grades of polypropylene.
Author Response
We thank the reviewer for his evaluation and constructive comments. In detail, the following adaptations have been made:
(1) The authors refer to a large number of Borstar® polymer technologies in the paper and hope to add some typical grades so that readers (especially engineers) can understand the differences between the different grades of polypropylene.
The grade names of some typical products of Borealis representative for the different types of polypropylene copolymers have been added.
(2) The authors could add some rheological data, especially the difference between single-distributed PP and bimodal cloth PP in shear rheology and capillary rheology tests.
This is probably a misunderstanding. In polypropylene grades used for fibre spinning and textile applications, only monomodal grades with rather narrow molecular weight distribution are applied. Regarding the difference in rheology between mono- and bimodal grades, …
(3) The authors should add some crystallization kinetic data such as activation energy, semi-crystallization time and other parameters to the section on crystallization properties to make a comprehensive addition.
We respectfully disagree, as such detail data would make this section more difficult to read. Figure 9 gives an overview, and a number of detail papers are referenced (this has been highlighted in the text).

Reviewer 3 Report
In this work, the authors reviewed the process technologies and catalyst structures, which is a good fit between engineering and chemistry. They discussed the development of Borstar® PP process and its capability in terms of wide product range and good economics. The manuscript is well prepared and the various sections were well organized; however, the following comment can help improve the manuscript.
· The novelty of the work is not clearly highlighted.
· Authors are recommended to use the same font and size for Tables as well as Figures.
· Please remove the grid lines in Figure 10.
· In SEM images, the scale bars need to be present more clearly.
Author Response
We thank the reviewer for his evaluation and constructive comments. In detail, the following adaptations have been made:
(1) The novelty of the work is not clearly highlighted.
Some comments in this respect have been added in introduction and conclusions.
(2) Authors are recommended to use the same font and size for Tables as well as Figures.
We respectfully disagree, as we think that Arial is a much better font for diagrams than Times.
(3) Please remove the grid lines in Figure 10.
This has been corrected.
(4) In SEM images, the scale bars need to be present more clearly.
The scale bars have been improved.
